# An herbal drug combination identified by knowledge graph alleviates the clinical symptoms of plasma cell mastitis patients: A nonrandomized controlled trial

**Caigang Liu[1]\*, Hong Yu[1†], Guanglei Chen[1†], Qichao Yang[2], Zichu Wang[2], Nan Niu[1], Ling Han[3], Dongyu Zhao[4], Manji Wang[5]\*, Yuanyuan Liu[2,6], Yongliang Yang[2]\***

[1]Cancer Stem Cell and Translation Medicine Lab, Innovative Cancer Drug Research and Development Engineering Center of Liaoning Province, Department of Oncology, Shengjing Hospital of China Medical University, Shenyang, China; [2]School of Bioengineering, Dalian University of Technology, Dalian, China; [3]National Engineering Research Center of Pharmaceutics of Traditional Chinese Medicine, China Resources Sanjiu Medical & Pharmaceutical Co., Ltd, Shenzhen, China; [4]International Cancer Institute, Peking University Health Science Center, Peking University, Beijing, China; [5]Shanghai BeautMed Corporation, Shanghai, China; [6]Department of Biology, University of Copenhagen, Copenhagen, Denmark

**\*For correspondence:**
angel-s205@163.com (CL);
147304368@qq.com (MW);
everbright99@foxmail.com (YY)

[†]These authors contributed
equally to this work

## Abstract

**Background:** Plasma cell mastitis (PCM) is a nonbacterial breast inflammation with severe and intense clinical manifestation, yet treatment methods for PCM are still rather limited. Although the mechanism of PCM remains unclear, mounting evidence suggests that the dysregulation of immune system is closely associated with the pathogenesis of PCM. Drug combinations or combination therapy could exert improved efficacy and reduced toxicity by hitting multiple discrete cellular targets.

**Methods:** We have developed a knowledge graph architecture toward immunotherapy and systematic immunity that consists of herbal drug–target interactions with a novel scoring system to select drug combinations based on target-hitting rates and phenotype relativeness. To this end, we employed this knowledge graph to identify an herbal drug combination for PCM and we subsequently evaluated the efficacy of the herbal drug combination in clinical trial.

**Results:** Our clinical data suggests that the herbal drug combination could significantly reduce the serum level of various inflammatory cytokines, downregulate serum IgA and IgG level, reduce the recurrence rate, and reverse the clinical symptoms of PCM patients with improvements in general health status.

**Conclusions:** In summary, we reported that an herbal drug combination identified by knowledge graph can alleviate the clinical symptoms of PCM patients. We demonstrated that the herbal drug combination holds great promise as an effective remedy for PCM, acting through the regulation of immunoinflammatory pathways and improvement of systematic immune level. In particular, the herbal drug combination could significantly reduce the recurrence rate of PCM, a major obstacle to PCM treatment. Our data suggests that the herbal drug combination is expected to feature prominently in future PCM treatment.

**Funding:** C. Liu's lab was supported by grants from the Public Health Science and Technology Project of Shenyang (grant: 22-321-32-18); Y. Yang's laboratory was supported by the National Natural Science Foundation of China (grant: 81874301), the Fundamental Research Funds for Central University (grant: DUT22YG122), and the Key Research project of 'be Recruited and be in Command' in Liaoning Province (2021JH1/10400050).
**Clinical trial number:** NCT05530226.

## Editor's evaluation

This study presents a valuable identification of an herbal drug combination for treating plasma cell mastitis (PCM), a breast inflammation with severe and intense clinical symptoms. The data were collected and analyzed using a solid approach in a clinical trial of 160 patients (NCT05530226). They can be used to understand how herbal drug combinations could help manage PCM patients.

## Introduction

Plasma cell mastitis (PCM) represents a serious inflammatory condition of breast that occurs in young and middle-aged females at nonpregnant and nonlactating period (*Cutler, 1949*). The main histopathological characteristics of PCM are the infiltration of plasma cells and lymphocytes in breast tissue (*Yu et al., 2012*). Interestingly, PCM shares similarity with breast cancer from the perspective of macroscopical or microscopical characteristics (*Cheng et al., 2015*). On the other hand, mounting evidence suggests that the dysregulation of immune system is closely associated with the pathogenesis of PCM. Recently, the incidence rate of PCM is quickly rising, yet the treatment methods for PCM are still rather limited. In clinical practice, surgical resection and hormone therapy remain two major treatments for PCM. Unfortunately, neither surgical resection nor hormone therapy could prevent recurrence of PCM. Moreover, the serious side effects of hormone therapy are still problematic (*Fleming et al., 2022*). Therefore, the discovery of effective PCM treatment or therapeutics with minimal side effects is clearly warranted.

Traditional Chinese Medicine (TCM) has a rather long history for the prevention and treatment of complex diseases in eastern Asia (*Tu, 2011*; *Tu, 2016*; *Yin et al., 2022*). Moreover, for decades, TCM has been often used as alternative or complementary medicine in the West. Indeed, Chinese herbal compounds have been successfully applied in the treatment of PCM in conjunction with Western medicine (*Zhang et al., 2020a*). In clinical practice, TCM refers to herbal entity prescription or formulae (also called 'Fangji'), which may exhibit coordinating or synergistic effects through the combination of multiple herb drugs (*Li et al., 2010*). However, the design of formulae in TCM is solely based on the principle of 'syndrome differentiation' according to the medicinal properties of herbal entities. Moreover, the molecular mechanisms for the 'formulae' in TCM remain rather elusive.

Knowledge graph has emerged as an advanced technology in the field of artificial intelligence that is able to connect entities in a graph based on their existing intricate relationships (*Ye et al., 2021*). In particular, knowledge graph can enable the rational design and identification of combination therapies for a specific disease or phenotypes (*Zeng et al., 2022*). Recently, we developed and constructed a knowledge graph for the discovery of herbal drug combination toward immunotherapy and systematic immunity. Subsequently, we identified a synergistic combination of herbal drugs for PCM via a scoring system based on target-hitting rates and phenotype relativeness. To verify our concept of design, we conducted a clinical trial experiment for the drug combination of herbal compounds mentioned above (ClinicalTrials.gov registration: NCT05530226). Strikingly, our clinical results demonstrated that the herbal drug combination identified by knowledge graph can markedly suppress various inflammatory cytokines in serum, restore clinical symptoms, and reduce the recurrence rates of PCM patients with improved global health status.

## Methods

### Construction of knowledge graph toward immunotherapy

We employed data mining techniques to collect and compile 240 targets of immunotherapy and systematic immunity from the PubMed database. Next, we collected and compiled 345 herbal drug entities officially released by the National Health Commission of China and the National Administration of Traditional Chinese Medicine. The intricate relations between the herbal drug entities and the immunotherapy targets were extracted from the PubMed database. These intricate relations were subjected to further manual curation. We used 13 ontology terms to describe the intricate relations (edges) in the knowledge graph. Moreover, 64 attributes of the medicinal properties for the herbal drug entities were collected and compiled from the Pharmacopoeia of China. Finally, we built the knowledge graph using Neo4j and Py2Neo tools that consists of 895 nodes and 2197 edges.

### Scoring system of the knowledge graph

To this end, we developed a scoring system to asses and predict synergistic drug combination of herbal drug entities (number of drugs, n) as follows:

$$\text{Score} = f(x) * g(x) * \sum_{i=1}^{x} \left\| \begin{pmatrix} h_1 \\ h_2 \\ h_3 \end{pmatrix} \right\| \tag{1}$$

Herein, $f(x)$ represents the penalty function. $f(x)$ value will be set to 0 if the medicinal properties of the drug combination fall into the contraindication rules in the Pharmacopoeia of China. Otherwise, $f(x)$ value will be set to 1. Herein, the penalty function is used to ensure that the herbal drug entities in the combination do not violate the contraindication rules in the Pharmacopoeia of China according to the medicinal attributes of herbal drugs.

$$g(x) = 1 - p_i \tag{2}$$

$g(x)$ is the target diversity function as above and $p_i$ is calculated as $p_i = w/t$ ; $t$ refers to the total number of targets within the knowledge graph, and $w$ refers to the total number of overlapping targets that the drug combination may hit. Hence, the target diversity function can be used as a measure to assess the diversity of the targets that the drug combination may hit. In other words, if each drug entity in the combination hits distinct targets, the $g(x)$ value will be set to 1. The last term of the scoring system is used as a measure to assess the relativeness of each drug entity in the combination and calculated as follows:

$$\sum_{i=1}^{x} \left\| \begin{pmatrix} h_1 \\ h_2 \\ h_3 \end{pmatrix} \right\| \tag{3}$$

In brief, $h_1$ represents the target-hitting rates of each drug entity in the combination and was calculated as follows, $h_1 = n_i/t$ ; $n_i$ is the number of hitting targets for each drug entity in the combination; again, $t$ refers to the total number of targets constituting the knowledge graph for the disease. It is noteworthy that the concept of hitting rates toward discrete targets has been used in the scoring function for the selection of synergistic drug combinations (*Jin et al., 2021*). $h_2$ represents the phenotype relativeness of each drug entity in the combination and $h_2 = c_2 * 1/x$, where $x$ is the number of drug entities in the combination and $c_2$ is the parameter, namely, if the drug entity is related to the phenotype of the disease (co-occurrence with the disease phenotype in the literature), then the $c_2$ value is set to 1; otherwise, the $c_2$ value is set to 0; $h_3$ represents the literature relativeness or confidence of each drug entity in the combination and calculated as follows:

$$h_3 = c_3 * \log \sqrt{\sum_{i=1}^{x} l \times (j + k)} \tag{4}$$

in which $l$ is the number of studies/publications that validated the association of drug entity with the specific disease (herein the knowledge graph refers to cancer immunotherapy), and $j$ and $k$ refer to

whether the relations of the drug entity with the disease have been validated in cell lines or patient (or animal) tissues, respectively. Namely, if the drug entity was validated in cancer cell lines or patient tissues, the $j$ or $k$ value will be set to 1, respectively. Otherwise, the $j$ or $k$ value will be set to 0; $c_3$ is the parameter and set to 1 here. Therefore, herein, a high score of $h_3$ implies that the drug combination is more relevant to cancer immunotherapy with high confidence of literature relativeness. Collectively, our scoring system can be used to select those drug combinations that are most relevant to disease phenotypes and those drug combinations that are able to hit most discrete targets related to immunotherapy.

## Design of the clinical trial

In brief, 160 female patients diagnosed with PCM at the Shengjing Hospital Affiliated to China Medical University were recruited in the clinical trial between January 2021 and February 2022. The patients were divided 1:1 into experimental group (EG) and control group (CG). It is noteworthy that, in order to demonstrate the therapeutic effects of TCM drug combination, we selected the patients who were treated with Western medicine in the real world during the same period. Therefore, the two groups of patients were divided into TCM treatment group (EG) and Western medicine treatment group (CG). There was no significant difference in baseline data such as age, body mass index, clinical classification, marriage, and child-bearing history between the two groups (*Supplementary file 1A*). The patients in the CG were orally treated with methylprednisolone tablets, 20 mg/day once a day. The patients in the EG were orally treated with 20 g/bag of herbal drug combination twice a day, once in the morning and once in the evening for 2 mo. The herbal drug combination was prepared as granules in the following formulae: Taraxacum 15 g, Fructus forsythiae 15 g, Honeysuckle 10 g, Uniflower swisscentaury root 8 g, Herba violae 20 g, Danshen 10 g, Astragalus 20 g, and Liquorice 8 g. The formula was determined by TCM experts on the basis of 'syndrome differentiation' as described in the Pharmacopoeia of China. Furthermore, the herbal drug combination in the form of granules was provided and prepared by the Shengjing Hospital Affiliated to China Medical University according to the standard requirement of clinical study by the National Medical Products Administration.

## Clinical trial protocol

The clinical trial for the herbal drug combination was registered at ClinicalTrials.gov and entitled 'A single arm study of Traditional Chinese Medicine for plasma cell mastitis" with registration code NCT05530226 (see *Figure 1—figure supplement 2*). The detailed clinical trial protocol has been provided as a separate document in '*Supplementary file 2*'.

## Measurement of serum inflammatory cytokines by ELSIA assay

Venous blood of the CG and EG was collected in a sterile non-anticoagulant test tube before and after treatment. The immune transmission turbidimetry was used according to the procedure of CRP kit, and automatic biochemical analyzer was used to detect the level of CRP. The levels of serum cytokines were measured by ELISA (Elabscience) according to the manufacturer's instructions.

## Measurement of serum immunoglobulin level

The venous blood of PCM patients in the two groups was collected in a sterile non-anticoagulant tube before and after treatment. The serum IgG and IgA were measured by rate scattering turbidimetry using Array 360 System automatic-specific protein analyzer (Beckman Company, USA).

## Assessment of clinical symptoms of PCM patients

The clinical symptoms were evaluated by an attending physician with board certification in pathology. The patients were scored before and after treatment according to the standard rating scale for PCM (*Supplementary file 1B*).

## Statistics

All data were evaluated as mean ± SEM. Statistical analysis of the quantitative multiple group comparisons was performed using ANOVA, followed by Tukey's test, whereas pairwise comparisons were performed using the *t*-test by GraphPad Prism 8 (GraphPad Software, La Jolla, CA). Results were considered to be statistically significant when $p<0.05$.

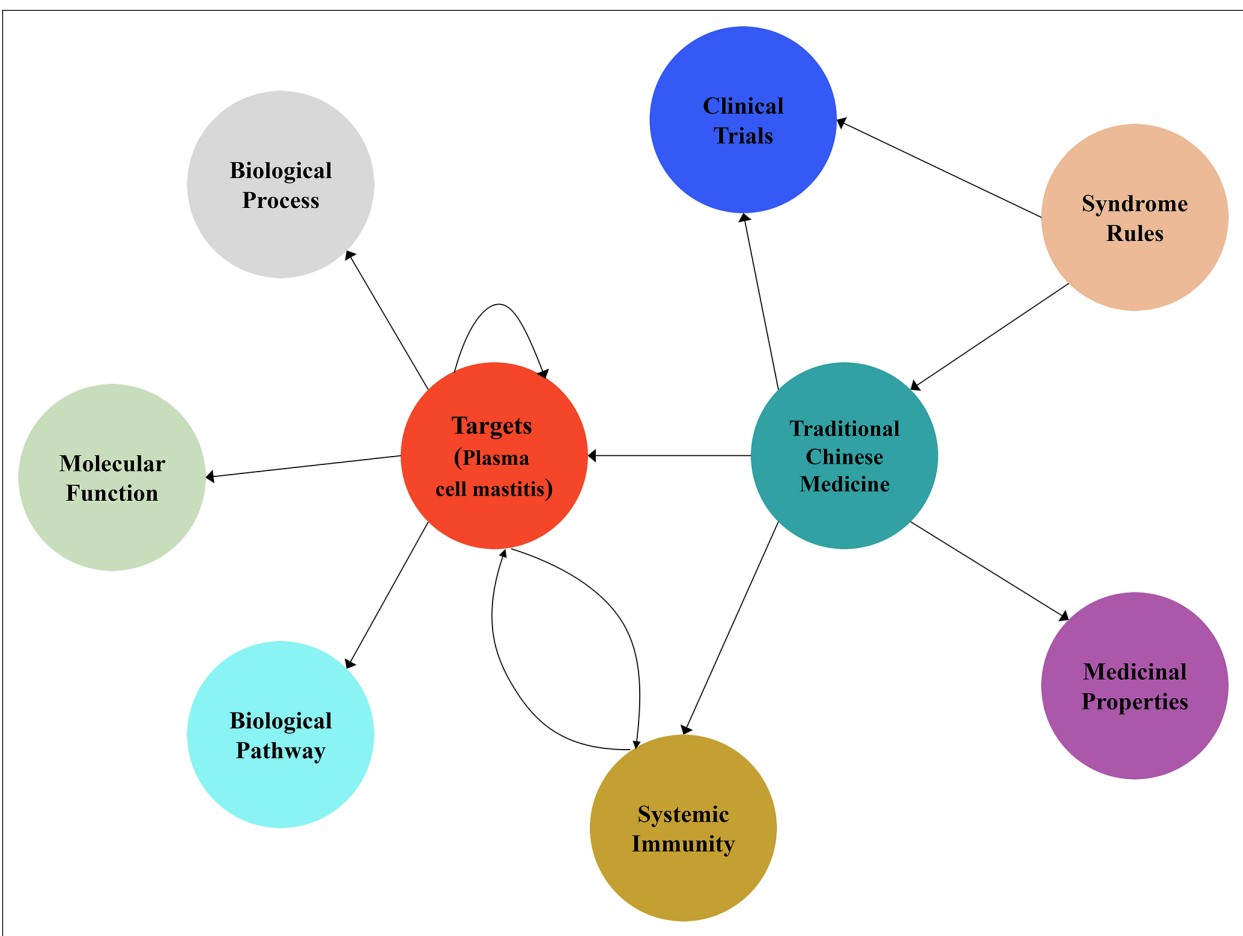

**Figure 1.** Schematic diagram of the knowledge graph for the drug discovery of plasma cell mastitis in the present work.

The online version of this article includes the following source data and figure supplement(s) for figure 1:

**Figure supplement 1.** Snapshot of the medical knowledge graph toward immunotherapy.

**Figure supplement 1—source data 1.** Full resolution file for *Figure 1—figure supplement 1*.

**Figure supplement 2.** CONSORT flowchart diagram of the clinical trial for the herbal drug combination.

## Results

In a previous study, we collected and compiled 240 targets for immunotherapy and systematic immunity from literature data (*Zhang et al., 2020b*). Recently, we collected 345 entities of herbal drugs documented in TCM books and herbal drugs announced by the National Administration of Traditional Chinese Medicine through advanced text-mining techniques. The existing intricate relationships between the herbal drugs and immunotherapy targets were also extracted and compiled via advanced text-mining techniques and manual curation for the construction of knowledge graph. We defined an ontology list consisting of 13 ontology terms describing the relations (edges) between herbal drug entities and the immunotherapy targets based on manual curation of literature data. Moreover, we collected the attributes of the medicinal properties of each herbal compound from the Pharmacopoeia of China (*Hao and Jiang, 2015*). We totally compiled and integrated 64 attributes of the medicinal properties of herbal drug entities into the knowledge graph. These medicinal properties are useful throughout the design of herbal drug combination. Finally, we built the knowledge graph using Neo4j and Py2Neo tools that consists of 895 nodes and 2197 edges (*Figure 1*, *Figure 1—figure supplement 1*), which can be visited online (http://www.ikgg.org/). Subsequently, we employed a scoring system (or so-called recommendation system) to asses and predict synergistic herbal drug combination from the knowledge graph. Note that the scoring function is able to identify those herbal drug combinations that are most related to specific phenotypes as well as herbal drug combinations

that are able to hit most discrete cellular targets, yet still following the principle of 'syndrome differentiation' as described in the Pharmacopoeia of China ('Methods'). Here, syndrome differentiation refers to the very basic principle of identifying and treating diseases in TCM. Syndrome (Zheng) is the presentation of the pathological changes during a specific disease course, including the location, cause, and nature of a disease. Moreover, 'Jun-Chen-Zuo-Shi' refers to the rules guided by syndrome differentiation to select multiple herbal drug entities to treat a specific disease in TCM. To this end, we used this scoring function to select herbal drug combinations consisting of eight herbal entities. We chose to identify drug combinations with eight entities because 'formulae' consisting of eight drugs are regarded as 'essence combination' in the TCM community. In short, we employed a combination generator that is able to randomly generate drug combinations with eight herbal drug entities for 10 rounds, each of which consists of 1000 random drug combinations. All the generated drug combinations from the 10 rounds were further ranked and evaluated. It is noteworthy that the scoring results of the 10 rounds are presented as normal distributions (*Figure 2—figure supplement 1*). The top 20 combinations from each round ranked by the scoring function were further curated and inspected by experts in TCM. Remarkably, we identified a specific drug combination that was ranked among the top 20 choices in all 10 rounds of calculation. The drug combination was chosen for further clinical study for two reasons. First, this drug combination was among the top 20 combinations in each round of our calculations. Second, we asked experts in TCM to inspect the top 20 combinations on the basis of 'syndrome differentiation' as described in the Pharmacopoeia of China, and finally the combination consisting of eight herbal drug entities, including 'Fructus forsythiae,' 'Herba violae,' 'Uniflower swisscentaury root,' 'Danshen,' 'Astragalus,' 'Taraxacum,' 'Liquorice,' and 'Honeysuckle,' was selected for further clinical study.

Next, we extracted the subgraph for the herbal drug combination mentioned above and created a network diagram for the drug combination using Cytoscape tools (*Reimand et al., 2019*; *Figure 2*). In total, the eight herbal drug entities in the combination regulate 46 cellular targets related to immunotherapy and systematic immunity such as HIF-1 (*Nan et al., 2022*), iNOS, IL-17, IL-6, IL-1β, mTOR, NLPR3, PD-L1, STAT3 (*Liu et al., 2020*), TGF-β, TLR2, and TLR4 (*Figure 2*). It is noteworthy that the medicinal properties of the eight drug entities could be classified into three major categories of 'heat-clearing and detoxicating,' 'Qi-tonifying,' and 'blood-activating menstruation regulating.' Moreover, we conducted pathway analysis for the herbal drug combination for plasma cell mastitis. Interestingly, we revealed that the herbal drug combination may modulate a few pathways related to systematic immunity, including 'Toll-like receptors cascades,' 'MAP kinase activation,' 'adaptive immune system,' 'growth hormone receptor signaling,' 'cytokine signaling in immune system,' and 'innate immune system' via Reactome Knowledgebase (*Jassal et al., 2020*; *Figure 3*). We believe that all these may account for the therapeutic profiles of the herbal drug combination toward PCM. For instance, 'Toll-like receptors cascades,' 'adaptive immune system,' 'cytokine signaling in immune system,' and 'innate immune system' are critical cellular pathways for systematic immunity that are directly associated with the pathogenesis of PCM. Moreover, the 'MAP kinase activation' pathway is associated with cellular defense and innate immunity, which are also crucial for the development and inflammatory conditions of PCM.

Efficacy was assessed every two cycles, and the results were summarized after 6 mo of treatment. The baseline characteristics are shown in *Supplementary file 1A*. Strikingly, our results demonstrated that a few inflammatory cytokines in the serum, including IL-2, IL-4, IL-6, IFN-γ, IL-1β, and TNF-α, were significantly downregulated in PCM patients after treatment of herbal drug combination compared to the CG treated with methylprednisolone (*Figure 4*). We chose to measure these cytokines in the experiments because they are often regarded as serum cytokine markers during the pathogenic development of PCM (*Liu et al., 2018*). In addition, we found that the serum levels of IgA and IgG were markedly suppressed in the treatment group of herbal drug combination compared to the CG (*Figure 5*). Note that both IgA and IgG have been found to be crucial diagnostic serum markers for PCM patients (*Xing et al., 2022*). Moreover, IgA is regarded as a major component of mucosal immunity that is closely related to the pathogenesis of PCM (*Betts et al., 2018*; *Bharathan and Mullarky, 2011*). Therefore, our data suggests that the herbal drug combination may enable the regulation of mucosal immunity and consequently downregulate IgA and IgG serum level. Furthermore, we conducted the standard Quality of Life questionnaire studies for PCM patients in the clinical experiment. Our results implicated that symptom score, pain score, and global health status of PCM

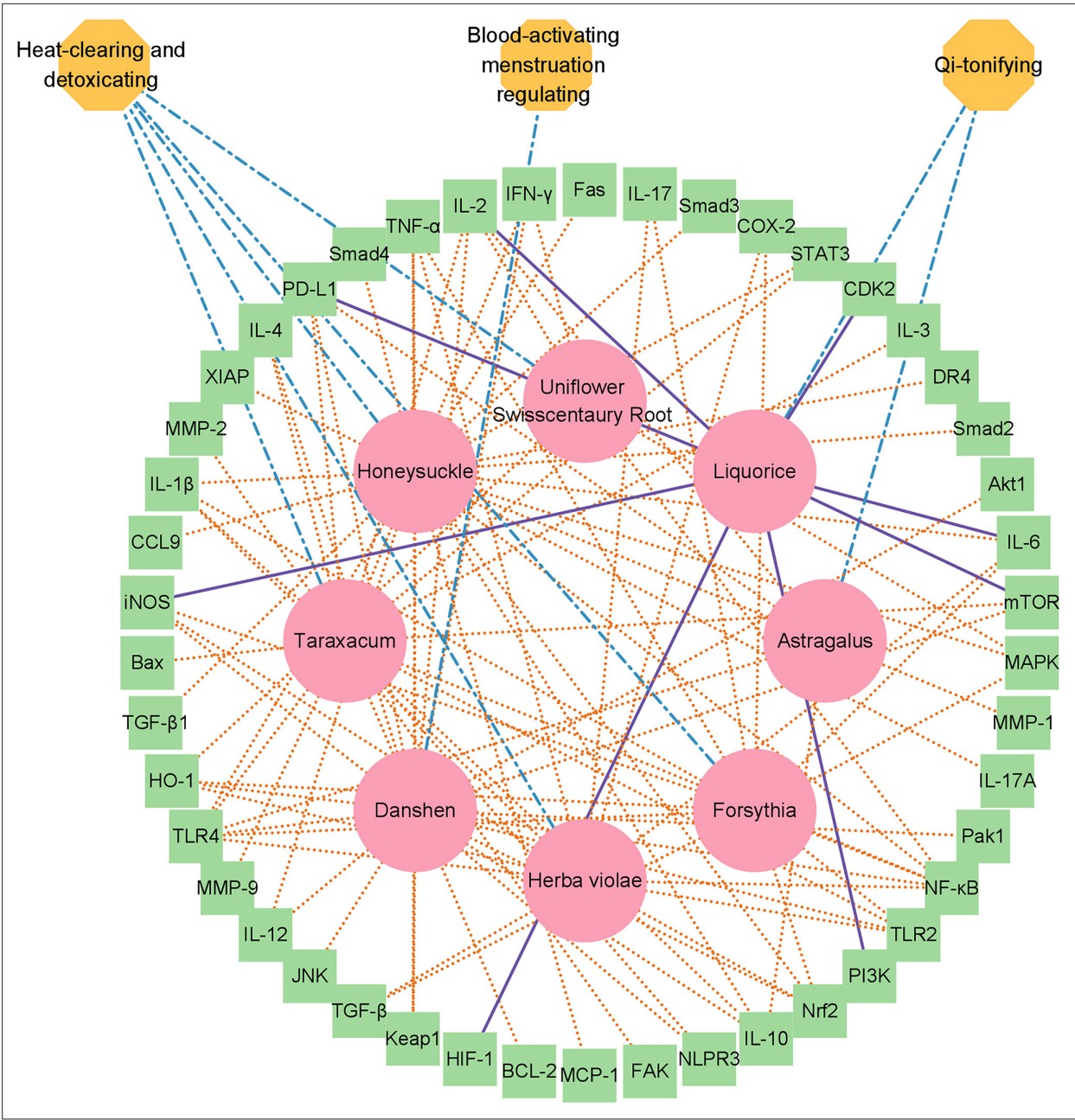

**Figure 2.** Network diagram of the herbal drug combination consisting of eight entities, including 'Honeysuckle,' 'Taraxacum,' 'Astragalus,' 'Danshen,' 'Forsythia,' 'Herba violae,' 'Liquorice,' and 'Uniflower swisscentaury root,' displayed in red circles. In total, 46 cellular targets related to systemic immunity were hit by the drug combination such as NLPR3, IL-17, TLR4, STAT3, IL-6, iNOS, and TLR2, which are displayed in the green box model. Three major medicinal attributes (properties) were identified for these herbal drug entities, including 'heat-clearing and detoxicating,' 'Qi-tonifying,' and 'blood-activating menstruation regulation'.

The online version of this article includes the following figure supplement(s) for figure 2:

**Figure supplement 1.** The score statistics of 10 rounds of random herbal drug combinations (1000 random combination from each round) for the treatment of plasma cell mastitis.

patients are significantly improved after treatment of the herbal drug combination compared to the CG (*Figure 6*). It is noteworthy that our results demonstrated that the recurrence rate of PCM patients in the treatment group was reduced to 3.75% compared to the recurrence rate of 12.5% in the CG (*Table 1*). Moreover, the incidence rate of adverse events of PCM patients in the treatment group was reduced to 6.25% compared to the recurrence rate of 11.25% in the CG (*Table 1*). In addition, we

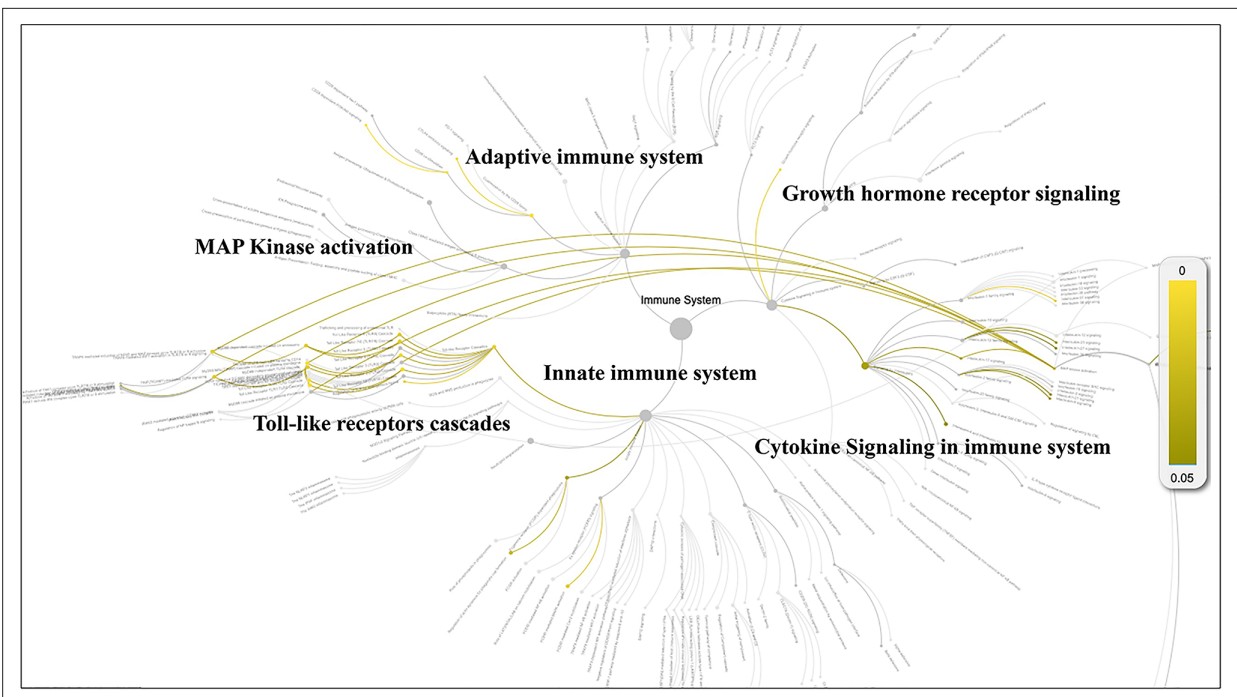

**Figure 3.** Pathway analysis for the potential cellular targets of the herbal drug combination via Reactome Knowledgebase. The statistically significant pathways are highlighted and displayed in yellow color (p<0.05).

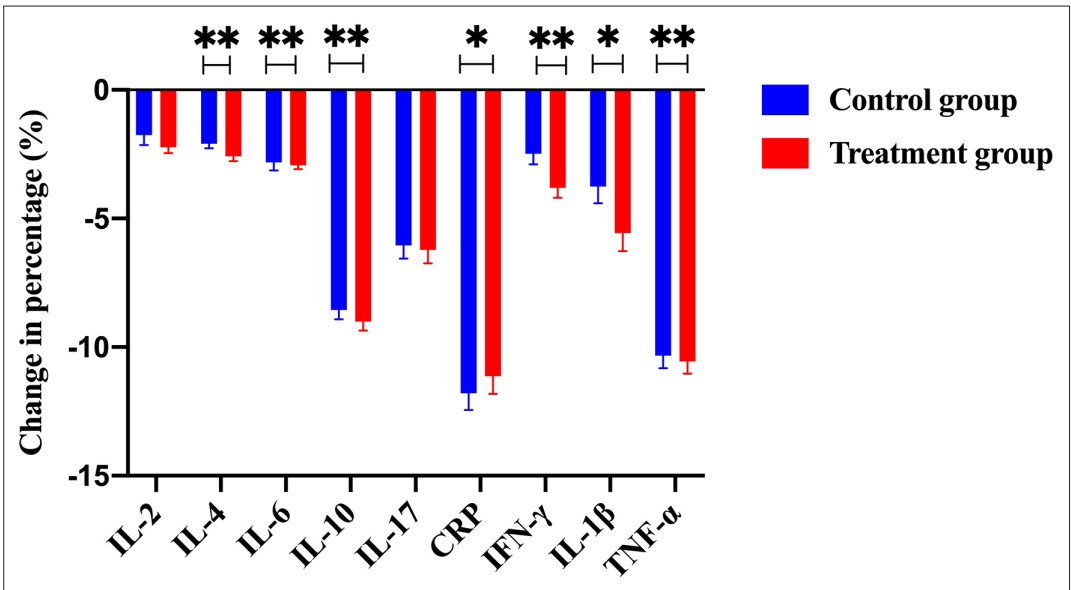

**Figure 4.** The change of percentage for numerous immunological cytokines, including IL-2, IL-4, IL-6, IL-10, IL-17, CRP, IFN-γ, IL-1β, and TNF-α, from the control group and the treatment group (with Traditional Chinese Medicine [TCM] treatment). Notably, a few key cytokines such as IL-2, IL-4, IFN-γ, IL-1β, and TNF-α were significantly downregulated in the treatment group compared to the control group.

The online version of this article includes the following source data for figure 4:

**Source data 1.** Patients' raw data of serum cytokines from the control group (without Traditional Chinese Medicine [TCM] treatment) and the treatment group (with TCM treatment) in the clinical trial (ClinicalTrials.gov: NCT05530226).

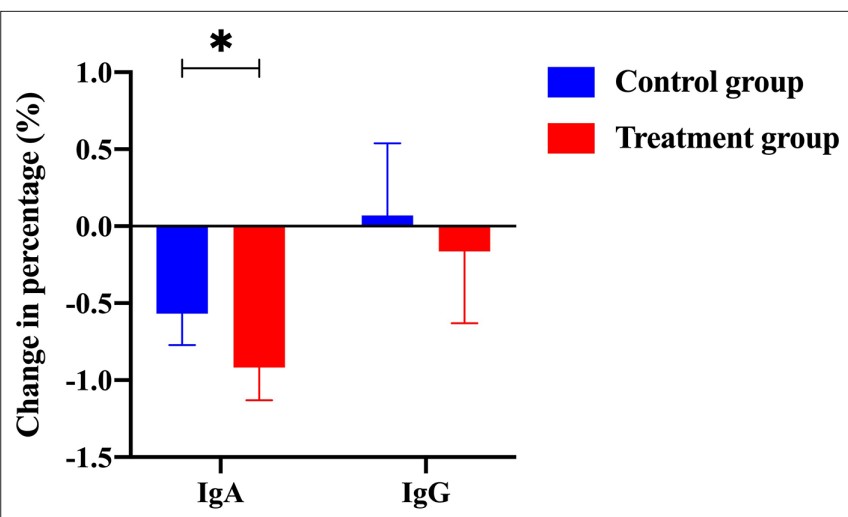

**Figure 5.** The change of percentage for IgA level and IgG level from the control group and the treatment group (with Traditional Chinese Medicine [TCM] treatment).

The online version of this article includes the following source data for figure 5:

**Source data 1.** Patients' raw data of serum IgA and IgG level from the control group (without Traditional Chinese Medicine [TCM] treatment) and the treatment group (with TCM treatment) in the clinical trial (ClinicalTrials.gov: NCT05530226).

observed that the clinical symptoms of PCM patients in the EG such as swelling, abscess, and fistula were reversed (*Figure 7*), *Supplementary file 3* after treatment of herbal drug combinations. The clinical symptom score in the EG is ~4.68 compared to the clinical symptom score of ~5.98 in the CG (*Table 2*). These results suggest that the herbal drug combination may achieve better efficacy for the treatment of PCM compared to methylprednisolone.

## Discussion

With the increasing amount of biomedical data, the traditional drug discovery campaign has been revolutionized with the aid of artificial intelligence techniques to accelerate the process and reduce the cost (*Yang et al., 2009*). In recent years, knowledge graph, a technique that can provide structured relations among entities and unstructured semantic relations associated with entities, has been introduced into the domain of drug discovery (*Zeng et al., 2022*). The advantage of employing knowledge graph for drug discovery lies in the capabilities of revealing structured associations between drug entities, cellular targets, biological pathways, and phenotypes for human disorders. This is useful for scientists to identify new indications or phenotypes for existing drugs or so-called drug repurposing. With the aid of scoring function or recommendation system, knowledge graph can also be used to design and identify drug combinations for a specific disease. Herein, for the first time, we introduced and employed the concept of knowledge graph to identify herbal drug combinations for the severe PCM with unmet medical needs.

Although the pathogenesis of PCM remains largely unclear, there have been numerous reports implicating that the overactivation of immunoinflammatory pathways plays an important role in the development of PCM (*Liu et al., 2015*). The major advantage of using TCM is that herbal drug combination can hit multiple discrete targets related to immunoinflammatory pathways with improved efficacy and reduced toxicity. Herein, for the first time, we showcase an example that identifies an herbal drug combination via knowledge graph toward PCM. In contrast to using the conventional principle of 'syndrome differentiation,' our knowledge graph consisting of intricate relations between herbal drug entities and immunotherapy targets coupled with scoring functions is able to automatically identify novel herbal drug combinations that can hit most discrete targets, making this strategy unique in the TCM community. This is because PCM is a rather complex disease that pathogenesis may involve multiple targets and immunoinflammatory pathways. Hence, we made the hypothesis

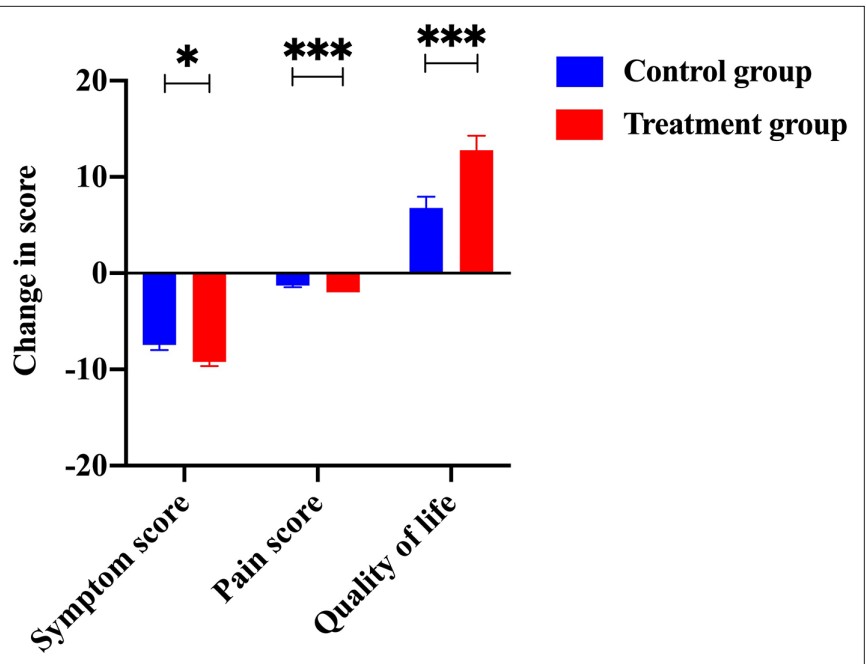

**Figure 6.** The change of scores from items, including symptom, pain, and living quality. The scores of symptoms, pain, and living quality from the treatment group were significantly improved compared to the control group.

The online version of this article includes the following source data and figure supplement(s) for figure 6:

**Source data 1.** Patients' raw data of symptom scores from the control group (without Traditional Chinese Medicine [TCM] treatment) and the treatment group (with TCM treatment) in the clinical trial (ClinicalTrials.gov: NCT05530226).

**Source data 2.** Patients' raw data of pain scores from the control group (without Traditional Chinese Medicine [TCM] treatment) and the treatment group (with TCM treatment) in the clinical trial (ClinicalTrials.gov: NCT05530226).

**Source data 3.** Patients' raw data of quality of life from the control group (without Traditional Chinese Medicine [TCM] treatment) and the treatment group (with TCM treatment) in the clinical trial (ClinicalTrials.gov: NCT05530226).

**Figure supplement 1.** The comparison of symptom score between different groups.

**Figure supplement 2.** The comparison of pain score between different groups.

**Figure supplement 3.** The comparison of score of life quality between different groups.

that drug combinations that can act on most discrete targets or pathways related to PCM might be more effective to alleviate the symptoms. Although we acknowledge that the inclusion of chemical ingredients from the herbal drugs may impact the outcome of our analysis and design, unfortunately, the inclusion of chemical ingredients in the knowledge graph is rather technically difficult due to the limited and incomplete datasets for the herbal drugs in the field of TCM. Nevertheless, our strategy captures the prominent feature of design for drug combinations toward a complex disease such as PCM. In the future, we plan to include multiple types of omics data such as genomic, transcriptomic,

**Table 1.** Comparison of operation rate, recurrence rate, and incidence of adverse reactions between the two groups (EG: experimental group; CG: control group).

| Groups | N (patients) | Operation | Recurrence | Incidence of adverse events |
|---|---|---|---|---|
| EG | 80 | 25 (31.25%) | 3 (3.75%) | 5 (6.25%) |
| CG | 80 | 47 (58.75%) | 10 (12.5%) | 9 (11.25%) |
| $X^2$ | | 12.22 | 4.103 | 1.252 |
| p | | <0.001 | 0.043 | 0.263 |

**Table 2.** Clinical symptom scores between the two groups in the clinical trial (EG: experimental group; CG: control group; The clinical symptom rating scale for plasma cell mastitis (PCM) is displayed in Supplementary file 1B).

| Groups | N (patients) | Baseline | Treated | t | p |
|---|---|---|---|---|---|
| EG | 80 | 13.90 ± 2.37 | 4.68 ± 3.40 | 22.19 | <0.001 |
| CG | 80 | 13.423 ± 2.70 | 5.98 ± 3.68 | 14.282 | <0.001 |
| t | | 1.189 | 2.323 | | |
| p | | 0.236 | 0.021 | | |

proteomic, metagenomic, or metabolomics data into the knowledge graph to reveal novel targets and enable novel drug discovery.

We want to remind the reader that a third arm (a placebo group) added to the clinical study might be useful to fully reveal the therapeutic effects of the herbal drug combination. Unfortunately, we

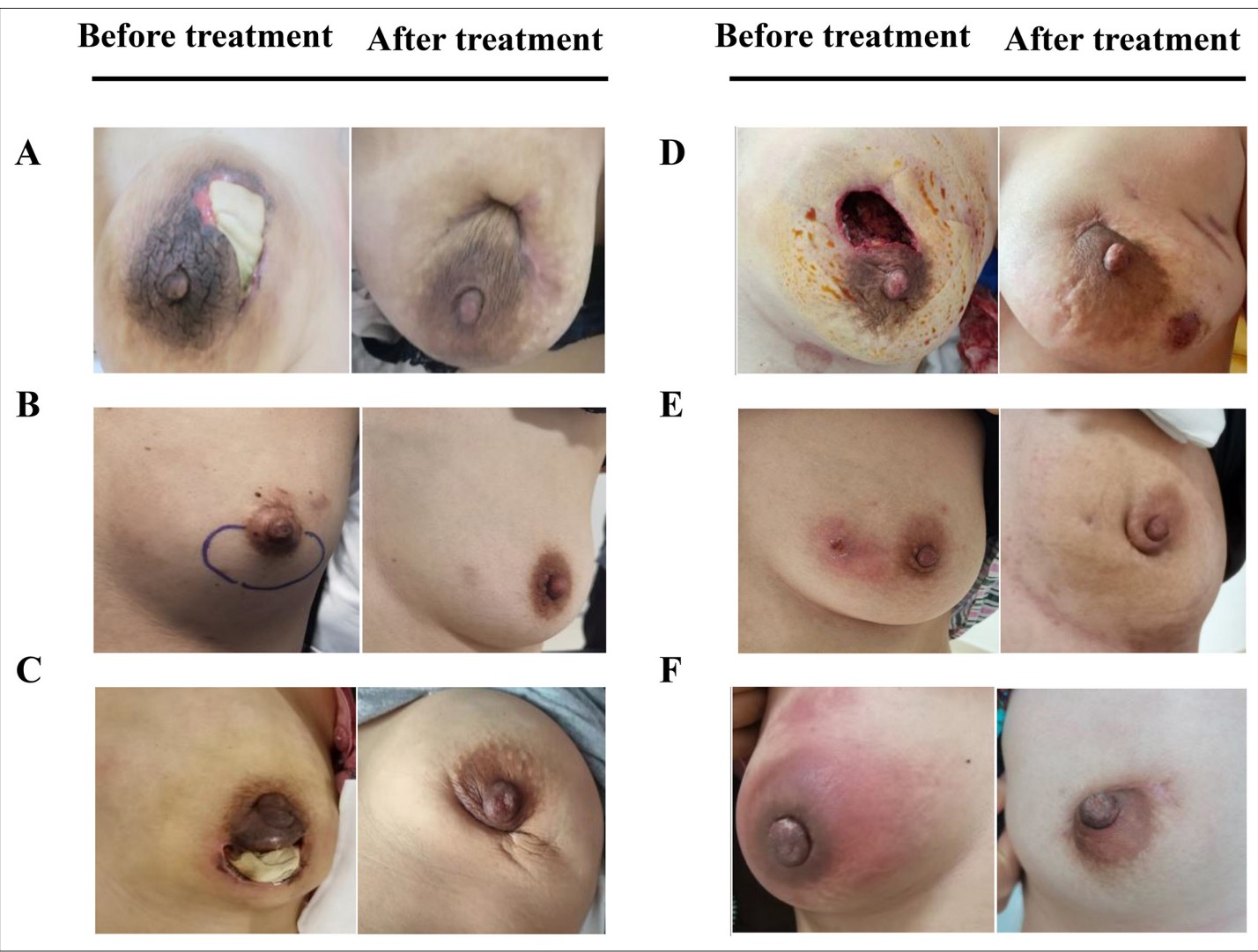

**Figure 7.** The comparison of whole breast for six representative plasma cell mastitis (PCM) patients in the experimental group (EG) before treatment and after treatment. (A-F), Six selected PCM patients in the clinical trial to reveal the effects of herbal drug combination. Written informed consents have been provided by all the patients (including written informed consent for the publication of the images in this figure), and detailed patient information is provided in *Supplementary file 3*.

were unable to add a third arm due to some ethical concerns. This is because PCM is a rather acute, severe, and intense clinical manifestation of breast. Without any treatment, the serious inflammatory condition of PCM may quickly advance into breast cancer. Therefore, this is a limitation of our clinical study, and we hope to design more delicate clinical trials in the future to demonstrate the therapeutic effects of herbal drug combination. Nevertheless, the detailed analysis of three indicators, including symptom score, pain score, and score of life quality, between the EG treated with herbal drug combination and the CG treated with methylprednisolone in the clinical study suggests that the herbal drug combination might be more effective in reversing the clinical conditions of PCM compared to the methylprednisolone treatment (see *Figure 6—figure supplement 1*, *Figure 6—figure supplement 2*, *Figure 6—figure supplement 3*).

In this study, our results revealed that the herbal drug combination identified by knowledge graph could suppress a few key immunoinflammatory cytokines, enhance the systematic immune levels, and significantly reduce the recurrence rates of PCM patients. Note that recurrence has become one major obstacle after surgical resection for PCM treatment in clinical practice. On the other hand, hormone therapy may increase the risk of side effects for PCM patients. Therefore, our approach of herbal drug combination may provide a new avenue for PCM treatment with less recurrence rate and reduced incidence rate of adverse events.

## Conclusion

In summary, we report the identification and clinical assessment of an herbal drug combination toward PCM. We demonstrated that the herbal drug combination holds great promise as an effective remedy for PCM, acting through the regulation of multiple cellular targets and immunoinflammatory pathways, which leads to the improvement in systematic immune level. In particular, the herbal drug combination could significantly reduce the recurrence rate of PCM, a major obstacle to PCM treatment. Our data suggests that the herbal drug combination is expected to feature prominently in the future PCM treatment. Moreover, these promising results underscore the potential of knowledge graph to identify drug combinations or other novel therapeutics across various types of human disorders.

## Ethical statement

The protocol was approved by the Institutional Review Board (IRB) of the China Medical University (approval number: 2021PS024T). This study was registered with ClinicalTrials.gov: NCT05530226. All patients provided written informed consent.

## Acknowledgements

Y Yang's laboratory was supported by the National Natural Science Foundation of China (grant: 81874301), the Fundamental Research Funds for Central University (grant: DUT22YG122), and the Key Research project of 'be Recruited and be in Command' in Liaoning Province (2021JH1/10400050); C Liu's lab was supported by grants from the National Natural Science Foundation of China (no. 81572609), China Medical University Major Construction Project (no. 2017ZDZX05), and Liaoning Colleges Innovative Talent Support Program (Cancer Stem Cell Origin and Biological Behavior).

## Additional information

### Competing interests

Caigang Liu: Senior editor, *eLife*. Ling Han: is an employee of China Resources Sanjiu Medical & Pharmaceutical. Manji Wang: is an employee of Shanghai BeautMed Corporation. Yongliang Yang: Reviewing editor, *eLife*. The other authors declare that no competing interests exist.

### Funding

| Funder | Grant reference number | Author |
| --- | --- | --- |
| National Natural Science Foundation of China | 81874301 | Yongliang Yang |

| Funder | Grant reference number | Author |
| --- | --- | --- |
| National Natural Science Foundation of China | 81572609 | Caigang Liu |
| Fundamental Research Funds for Central University | DUT22YG122 | Yongliang Yang |
| Key Research project of 'be Recruited and be in Command' in Liaoning Province | 2021JH1/10400050 | Yongliang Yang |
| China Medical University Major Construction Project | 2017ZDZX05 | Caigang Liu |
| Liaoning Colleges Innovative Talent Support Program | Cancer Stem Cell Origin and Biological Behavior | Caigang Liu |

The funders had no role in study design, data collection and interpretation, or the decision to submit the work for publication.

### Author contributions

Caigang Liu, Conceptualization, Resources, Supervision, Funding acquisition, Project administration; Hong Yu, Data curation, Supervision, Investigation, Project administration; Guanglei Chen, Validation, Visualization, Project administration; Qichao Yang, Data curation, Software, Visualization, Methodology; Zichu Wang, Software, Methodology; Nan Niu, Validation, Project administration; Ling Han, Resources, Validation, Visualization; Dongyu Zhao, Manji Wang, Validation, Investigation, Visualization; Yuanyuan Liu, Data curation, Software; Yongliang Yang, Conceptualization, Resources, Supervision, Funding acquisition, Investigation, Methodology, Writing - original draft, Project administration, Writing - review and editing

### Author ORCIDs

Caigang Liu http://orcid.org/0000-0003-2083-235X
Hong Yu http://orcid.org/0000-0002-4557-1558
Yongliang Yang http://orcid.org/0000-0003-0449-0599

### Ethics

Clinical trial registration ClinicalTrials.gov: NCT05530226.
Human subjects: The protocol was approved by the Institutional Review Board (IRB) of the China Medical University (approval number: 2021PS024T). This study was registered with https://clinical-trials.gov/: NCT05530226. All patients provided written informed consent. Patients also provided written informed consent for the publication of the images in Figure 7.

### Decision letter and Author response

Decision letter https://doi.org/10.7554/eLife.84414.sa1
Author response https://doi.org/10.7554/eLife.84414.sa2

---

# Additional files

### Supplementary files

• Supplementary file 1. Baseline charateristics and rating scale for the clinical trial. (A) Baseline characteristics of patients in the clinical trial. (B) Clinical symptom rating scale for plasma cell mastitis (PCM).

• Supplementary file 2. Clinical protocol for the clinical trial study.

• Supplementary file 3. Some detailed information for the six patients displayed in *Figure 7*.

• MDAR checklist

• Source code 1. Python source code for the scoring system of knowledge graph to assess and select appropriate drug combinations.

## Data availability

Figure 1–3 are computational and therefore no data have been generated. In addition, Figure 4—source data 1, Figure 5—source data 1, Figure 6—source data 1, Figure 6—source data 2 and Figure 6—source data 3 contain the numerical data used to generate the figures.

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
