## [Editor Report]

This study presents a valuable identification of an herbal drug combination for treating plasma cell mastitis (PCM), a breast inflammation with severe and intense clinical symptoms. The data were collected and analyzed using a solid approach in a clinical trial of 160 patients (NCT05530226). They can be used to understand how herbal drug combinations could help manage PCM patients.

---

## [Decision Letter]

**Decision letter after peer review:**

Thank you for submitting your article "A herbal drug combination identified by knowledge graph alleviates the clinical symptoms of plasma cell mastitis patients: a non-randomized controlled trial" for consideration by *eLife*. Your article has been reviewed by 3 peer reviewers, one of whom is a member of our Board of Reviewing Editors, and the evaluation has been overseen by Mone Zaidi as the Senior Editor. The reviewers have opted to remain anonymous.

Essential revisions:

(1) Please address Reviewer 1's comments about the patient groups and confounding factors as well as expand the rationale for the evaluation of the clinical trial as noted for example, by reviewer 2's comments about the combination drug score.

(2) Please address the relatively minor comments of all three reviewers.

*Reviewer #1 (Recommendations for the authors):*

Figure 2 looks like it could be moved to supplementary data.

Figure 3 text was too small to read.

The presented changes in figures 5 and 6, while statistically significant do not demonstrate significant difference. Perhaps a third, placebo, arm is needed whereby the steroids and the herbal groups induce immune changes not seen in the placebo group? That data would be clear and convincing, but as presented it is difficult to understand if this is a medication effect o=r the natural course of disease in these patients.

*Reviewer #2 (Recommendations for the authors):*

Below are some specific points for the manuscript,

1. It would be necessary to elaborate the advantages of employing knowledge graph to design drug combinations for a specific human disease in the Discussion section of the manuscript.

2. In Figure 4, the authors claimed that 'a few pathways related to systematic immunity' were identified. In page 8, line 158, the authors should discuss in length for the specific roles of these pathways in PCM pathogenesis.

3. The drug combination in the clinical trial didn't obtain the highest score (Figure 2). The authors need to explain this in the Discussion section.

4. Figure 1, Supplement 1, the authors need to explain the color scheme (pink, yellow and green etc) for the entities.

5. Page 6, line 133, please explain and specify the term 'syndrome differentiation'. The reader who are not in TCM domain may not understand the term.

6. In page 6, the authors claimed that the scoring system is able to identify 'herbal drug combinations that are able to hit most discrete cellular targets'. The authors need to explain and elaborate why this criteria is crucial for the identification of drug combinations for PCM in the Discussion section.

7. The authors need to provide more details for their clinical study. For example, more details are required for the patients in Figure 6.

*Reviewer #3 (Recommendations for the authors):*

This is a rather interesting study that introduced a new concept in the domain of TCM and herbal drugs. Below are some specific suggestions which I think may help to improve the manuscript,

1. The herbal drug combination used in the clinical study didn't obtain the highest score during the scoring process (Figure 2). The authors need to explain and elaborate this in a paragraph.

2. In Figure 3, the network diagram analysis from the knowledge graph revealed that NLPR3, IL-17, TLR4, STAT3, IL-6, iNOS and TLR2 are possible cellular targets for the herbal drug combination. The authors need to elaborate or provide some references that NLPR3, IL-17, TLR4, STAT3, IL-6, iNOS and TLR2 are relevant to PCM pathogenesis.

3. The authors need to explain and elaborate 'penalty function', in page 14.

4. Line 326, the authors mean 'validated via cellular model'?

5. Line 329, the author refers to 'systematic immunity'?

6. Page 17, line 347, 'The herbal drug combination was prepared as granules in the following formulae', The authors need to explain how this formula was determined.

7. In Figure 8, the authors need to provide more details for the clinical pictures in a paragraph. For example, age, treatment period and brief descriptions of the symptoms etc.

8. The authors need to explain why the drug combination consisting of eight herbal drugs is selected in their study. Page 7, line 137, the authors made a statement that 'eight drugs are regarded as essence combination'. The authors need to explain and elaborate this point in the manuscript.

9. Is it feasible that the authors could include chemical ingredients of the herbal drugs into their knowledge graph for the identification of herbal drug combination? The authors should elaborate more for this point in the manuscript.

---

## [Author Response]

Reviewer #1 (Recommendations for the authors):Figure 2 looks like it could be moved to supplementary data.

This has been fixed. Per the referee’s suggestion, we have move Figure 2 to supplementary data.

Figure 3 text was too small to read.

This has been fixed. Per the referee’s comment, we have rearranged and enlarged the text in Figure 3 (now Figure 2 in draft-track-change). Please see draft-track-change for more details.

The presented changes in figures 5 and 6, while statistically significant do not demonstrate significant difference. Perhaps a third, placebo, arm is needed whereby the steroids and the herbal groups induce immune changes not seen in the placebo group? That data would be clear and convincing, but as presented it is difficult to understand if this is a medication effect o=r the natural course of disease in these patients.

We want to thank the referee for this very insightful and expert comment. We fully agree with the referee that a third arm (placebo group) added in the clinical trial may be useful to further reveal the therapeutic effects of herbal drug combinations. Indeed, we met some difficulties to add the placebo group in the clinical trial and unfortunately, we were unable to obtain the ethnical approval from the Research Ethnical Committee of our university should we want to add the placebo group in the trial. This is because plasma cell mastitis (PCM) is a rather acute, severe and intense clinical manifestation of breast. Without any treatment, the serious inflammatory condition of PCM may quickly advance into breast cancer. In addition, the patients may feel tremendous pain without treatment due to the quick development of PCM. Therefore, the Research Ethnical Committee raised concerns about the placebo group and we were unable to add the placebo group in the trial. Nevertheless, we fully agree with the referee that a third arm could be useful and we elaborate this limitation of our clinical study in a paragraph in the Discussion section of our manuscript. Moreover, we conducted additional analysis of the clinical data. The detailed analysis of three indicators including symptom score, pain score and score of life quality between the experiment group treated with herbal drug combination and the control group treated with methylprednisolone in the clinical study suggest that the herbal drug combination might be more effective to reverse the clinical conditions of PCM as compared to the methylprednisolone treatment (see Figure 6 —figure supplement 1-2-3). Please see below or draft-track-change, line 234-246 for more details.

“We want to remind the reader that a third arm (a placebo group) added in the clinical study might be useful to fully reveal the therapeutic effects of the herbal drug combination. Unfortunately, we were unable to add a third arm due to some ethnical concerns. This is because plasma cell mastitis (PCM) is a rather acute, severe and intense clinical manifestation of breast. Without any treatment, the serious inflammatory condition of PCM may quickly advance into breast cancer. Therefore, this is a limitation of our clinical study and we hope to design more delicate clinical trial in the future to demonstrate the therapeutic effects of herbal drug combination. Nevertheless, the detailed analysis of three indicators including symptom score, pain score and score of life quality between the experiment group treated with herbal drug combination and the control group treated with methylprednisolone in the clinical study suggest that the herbal drug combination might be more effective to reverse the clinical conditions of PCM as compared to the methylprednisolone treatment (see Figure 6 —figure supplement 1-2-3).”

Reviewer #2 (Recommendations for the authors):Below are some specific points for the manuscript,1. It would be necessary to elaborate the advantages of employing knowledge graph to design drug combinations for a specific human disease in the Discussion section of the manuscript.

Many thanks for this critical and helpful comment. Per the referee’s suggestion, we have elaborated the advantage of the knowledge graph towards the design of drug combinations in the Discussion section of the manuscript. Please see below or draft-track-change, line 214-222 for more details.

“The advantage of employing knowledge graph for drug discovery lies in the capabilities of revealing structured associations between drug entities, cellular targets, biological pathways and phenotypes for human disorders. This is useful for scientists to identify new indications or phenotypes for existing drugs, or so-called drug repurposing. With the aid of scoring function or recommendation system, knowledge graph can also be used to design and identify drug combinations for a specific disease. Herein, for the first time, we introduced and employed the concept of knowledge graph to identify herbal drug combinations for the severe Plasma cell mastitis (PCM) with unmet medical needs.”

2. In Figure 4, the authors claimed that 'a few pathways related to systematic immunity' were identified. In page 8, line 158, the authors should discuss in length for the specific roles of these pathways in PCM pathogenesis.

We want to thank the referee for this insightful comment. This has been fixed. Per the referee’s suggestion, we have discussed in length for the specific roles of the related pathways in PCM. Please see below or draft-track-change, line 214-222 for more details.

“For instance, ‘Toll-like receptors cascades’, ‘Adaptive immune system’, ‘Cytokine signaling in immune system’ and ‘Innate immune system’ are critical cellular pathways for systematic immunity which are directly associated with the pathogenesis of PCM. Moreover, the ‘MAP kinase activation’ pathway is associated with cellular defense and innate immunity which are also crucial for the development and inflammatory conditions of PCM.”

3. The drug combination in the clinical trial didn't obtain the highest score (Figure 2). The authors need to explain this in the Discussion section.

Thanks for this expert and helpful comment. This has been fixed. The drug combination was chosen for the further clinical study for two reasons. First, this drug combination was among top twenty combinations in each round of calculations. Second, we asked experts in TCM to inspect the top twenty combinations on the basis of ‘syndrome differentiation’ as described in Pharmacopoeia of China and finally the combination consisting of eight herbal drug entities including ‘Fructus forsythiae’, ‘Herba violae’, ‘Uniflower swisscentaury root’, ‘Danshen’, ‘Astragalus’, ‘Taraxacum’, ‘Liquorice’ and ‘Honeysuckle’ was selected for further clinical study. Please see below or draft-track-change, line 144-151 for more details.

“The drug combination was chosen for the further clinical study for two reasons. First, this drug combination was among top twenty combinations in each round of our calculations. Second, we asked experts in TCM to inspect the top twenty combinations on the basis of ‘syndrome differentiation’ as described in Pharmacopoeia of China and finally the combination consisting of eight herbal drug entities including ‘Fructus forsythiae’, ‘Herba violae’, ‘Uniflower swisscentaury root’, ‘Danshen’, ‘Astragalus’, ‘Taraxacum’, ‘Liquorice’ and ‘Honeysuckle’ was selected for further clinical study (see Figure 2).”

4. Figure 1, Supplement 1, the authors need to explain the color scheme (pink, yellow and green etc) for the entities.

Thank you for this critical comment. This has been fixed. We have explained the various color scheme in the figure legend. Please see page 22, in draft-track-change for more details.

5. Page 6, line 133, please explain and specify the term 'syndrome differentiation'. The reader who are not in TCM domain may not understand the term.

We are grateful for this critical comment. This has been fixed. Per the referee’s suggestion, we have explained the term ‘syndrome differentiation’ in the manuscript. Please see Please see below or line 134-138, in draft-track-change for more details.

“Here, syndrome differentiation refers to the very basic principle of identifying and treating disease in TCM. Syndrome (Zheng) is the presentation of the pathological changes during a specific disease course including the location, cause and nature of a disease. Moreover, ‘Jun-Chen-Zuo-Shi’ refers to the rules guided by syndrome differentiation to select multiple herbal drug entities to treat a specific disease in TCM.”

6. In page 6, the authors claimed that the scoring system is able to identify 'herbal drug combinations that are able to hit most discrete cellular targets'. The authors need to explain and elaborate why this criteria is crucial for the identification of drug combinations for PCM in the Discussion section.

Many thanks for this critical comment. Per the referee’s suggestion, we have explained why the criterion of the scoring system is crucial for the identification of drug combinations in the manuscript. Please see below or line 249-252 in draft-track-change for more details.

“This is because PCM is a rather complex disease which pathogenesis may involve multiple targets and immunoinflammatory pathways. Hence, we made the hypothesis that drug combinations that can act on most discrete targets or pathways related to PCM might be more effective.”

7. The authors need to provide more details for their clinical study. For example, more details are required for the patients in Figure 6.

This has been fixed. We have provided detailed information for the six patients in Figure 6 as supplemental materials (SI_patients_information).

Reviewer #3 (Recommendations for the authors):This is a rather interesting study that introduced a new concept in the domain of TCM and herbal drugs. Below are some specific suggestions which I think may help to improve the manuscript,1. The herbal drug combination used in the clinical study didn't obtain the highest score during the scoring process (Figure 2). The authors need to explain and elaborate this in a paragraph.

Thanks for this expert and helpful comment. This has been fixed. The drug combination was chosen for the further clinical study for two reasons. First, this drug combination was among top twenty combinations in each round of calculations. Second, we asked experts in TCM to inspect the top twenty combinations on the basis of ‘syndrome differentiation’ as described in Pharmacopoeia of China and finally the combination consisting of eight herbal drug entities including ‘Fructus forsythiae’, ‘Herba violae’, ‘Uniflower swisscentaury root’, ‘Danshen’, ‘Astragalus’, ‘Taraxacum’, ‘Liquorice’ and ‘Honeysuckle’ was selected for further clinical study. Please see below or draft-track-change, line 144-151 for more details.

“The drug combination was chosen for the further clinical study for two reasons. First, this drug combination was among top twenty combinations in each round of our calculations. Second, we asked experts in TCM to inspect the top twenty combinations on the basis of ‘syndrome differentiation’ as described in Pharmacopoeia of China and finally the combination consisting of eight herbal drug entities including ‘Fructus forsythiae’, ‘Herba violae’, ‘Uniflower swisscentaury root’, ‘Danshen’, ‘Astragalus’, ‘Taraxacum’, ‘Liquorice’ and ‘Honeysuckle’ was selected for further clinical study (see Figure 2).”

2. In Figure 3, the network diagram analysis from the knowledge graph revealed that NLPR3, IL-17, TLR4, STAT3, IL-6, iNOS and TLR2 are possible cellular targets for the herbal drug combination. The authors need to elaborate or provide some references that NLPR3, IL-17, TLR4, STAT3, IL-6, iNOS and TLR2 are relevant to PCM pathogenesis.

We are grateful for this critical and helpful comment. This has been fixed. We have provided references for these putative targets. Please see page 8, line 161-162 in draft-track-change for more details.

3. The authors need to explain and elaborate 'penalty function', in page 14.

This has been fixed. We have explained the ‘penalty function’ in the manuscript. Please see below or page 13, line 339-342 in draft-track-change for more details.

“Herein, the penalty function is used to ensure that the herbal drug entities in the combination didn’t violate the contraindication rules in Pharmacopoeia of China according to the medicinal attributes of herbal drugs.”

4. Line 326, the authors mean 'validated via cellular model'?

This has been fixed. Here we mean ‘validated via cellular model’.

5. Line 329, the author refers to 'systematic immunity'?

This has been fixed. Herein we refer to 'systematic immunity'.

6. Page 17, line 347, 'The herbal drug combination was prepared as granules in the following formulae', The authors need to explain how this formula was determined.

This has been explained. Please see below or page 19, line 393-398 in draft-track-change for more details.

“The formula was determined by TCM experts on the basis of ‘syndrome differentiation’ as described in Pharmacopoeia of China. Furthermore, the herbal drug combination in the form of granules was provided and prepared by Shengjing Hospital Affiliated to China Medical University according to the standard requirement of clinical study by National Medical Products Administration (NMPA).”

7. In Figure 8, the authors need to provide more details for the clinical pictures in a paragraph. For example, age, treatment period and brief descriptions of the symptoms etc.

This has been fixed. We have provided detailed information for the six patients in Figure 6 as supplemental materials (SI_patients_information).

8. The authors need to explain why the drug combination consisting of eight herbal drugs is selected in their study. Page 7, line 137, the authors made a statement that 'eight drugs are regarded as essence combination'. The authors need to explain and elaborate this point in the manuscript.

We are thankful for this critical and helpful comment. This has been fixed. We chose to identify drug combinations with eight entities because ‘formulae’ consisting of eight drugs are regarded as ‘essence combination’ in TCM field. Please see page 7, line 140-142 in draft-track-change for more details.

9. Is it feasible that the authors could include chemical ingredients of the herbal drugs into their knowledge graph for the identification of herbal drug combination? The authors should elaborate more for this point in the manuscript.

Many thanks for this insightful and helpful comment. We fully agree with the referee that the inclusion of chemical ingredients from the herbal drugs may impact the outcome of our analysis and design. Unfortunately, the inclusion of chemical ingredients in the knowledge graph is rather technically difficult due to the limited and incomplete datasets for the herbal drugs in the field of TCM. Nevertheless, our strategy captures the prominent feature of design for drug combinations towards a complex disease such as PCM. In the future, we plan to include multiple types of omics data such as genomic, transcriptomic, proteomic, metagenomic or metabolomics data into the knowledge graph to reveal novel targets and enable novel drug discovery. Please see page 12, line 252-260 in draft-track-change for more details. We hope this could clarify the concerns.